# BASEL: TARGET-AWARE BASIS SELECTION FOR LANGUAGE MODELS

## ABSTRACT

As the size of language models increases, they deliver substantial performance improvements across a variety of applications. However, this growth also leads to greater computational demands, making deployment on resource-constrained devices—such as personal computers and mobile or wearable devices—more challenging, and significantly raising inference costs on cloud servers. To address these challenges, we introduce a method to streamline language models. We observe that language models pretrained on general datasets often include redundant components that are unnecessary for particular tasks. Our approach identifies and removes these redundant parts, retaining only the essential components for the intended applications. Specifically, we represent the weight matrices of language models as a linear combination of base components, eliminate the irrelevant bases, and introduce new bases that enhance performance for target tasks. Evaluations show that our method reduces model size much more significantly—by up to 1.7 times—while maintaining similar accuracy, compared to state-of-the-art techniques, across a range of applications.

## 1 INTRODUCTION

Large language models (LLMs) have significantly enhanced the performance of various applications in natural language processing, computer vision, and beyond. However, their large model sizes pose a bottleneck for many practical uses. The substantial computing resources required for LLM inference make it challenging to deploy them on devices with limited capabilities, such as personal computers and mobile/wearable devices. Moreover, even on hardware platforms with ample computing power, deploying LLMs consumes a significant amount of energy, raising concerns about sustainability. Therefore, it is essential to reduce the size of LLMs after pretraining to ease their computational demands and lower energy consumption.

Our approach exploits the relationship between pretrained models and specific target applications. Large language models (LLMs) are typically pretrained on vast datasets encompassing a wide range of tasks, many of which share common characteristics. This shared pretraining fosters synergies that enhance the performance of LLMs. However, the diversity among these tasks also introduces redundancy into the models. As demonstrated by our interpretation results in Section 3 shows, LLMs contain a significant number of redundant components that are unnecessary for a specific target application. By removing these redundant parts and retaining only the relevant ones, we can reduce the model's size while preserving its performance on the target application. Since many scenarios only require support for a specific type of application, this approach effectively lowers the computing resource requirements and reduces inference costs.

However, how do we identify the beneficial and redundant components of LLMs for a specific application? In this work, we address this problem through the lens of matrix factorization. Singular Value Decomposition (SVD) (Golub & Van Loan, 1996) factorizes a weight matrix $\mathbf{W}$ into the product of three matrices $\mathbf{U}$, $\mathbf{S}$, and $\mathbf{V}$, i.e., $\mathbf{W} = \mathbf{U}\mathbf{S}\mathbf{V}^{\mathrm{T}} = \sum_i s_i \mathbf{u}_i \mathbf{v}_i^{\mathrm{T}}$, where $s_i$ are (positive) singular values, and $\mathbf{u_i}$ and $\mathbf{v_i}$ are column vectors of $\mathbf{U}$ and $\mathbf{V}$ with unit norms. Our interpretation results show that these column vectors $\mathbf{u_i}$ and $\mathbf{v_i}$ may carry specific meanings. For instance, in the LLaMA 2-7B model (Touvron et al., 2023), when factorizing the weight matrix $\mathbf{W_O^h W_V^h}$ of an attention head that is likely useful for code generation tasks, de-embedding the resulting column vectors $\mathbf{u_i}$ and $\mathbf{v_i}$ reveals tokens like _in, <0x0A>, _and, _to, and _for. These column vectors

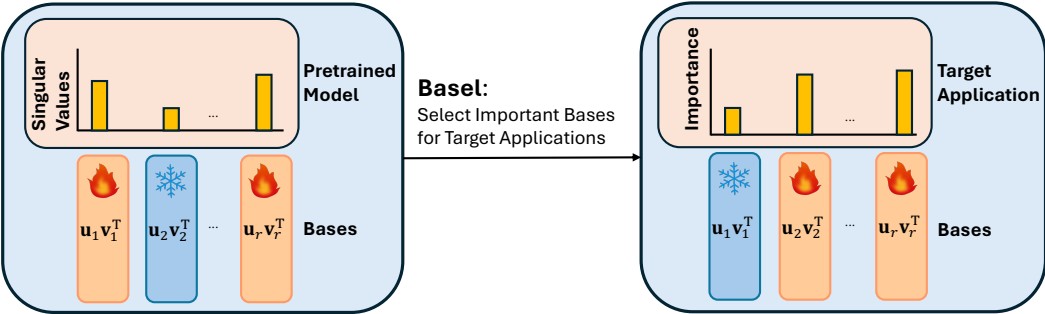

Figure 1: Basel: Identify and select the important bases for target applications during compression.

are evidently highly useful for code generation but may be less relevant for tasks such as mathematical reasoning.

Inspired by this observation, we propose *Basel*, a low-rank decomposition approach to effectively compress LLMs for target applications. Figure 1 illustrates the key idea of Basel. We view each weight matrix in LLMs as a linear combination of bases $\mathbf{u_i}\mathbf{v_i}^T$ with singular values $s_i$ as their weights. These bases are valuable representations stored in the pretrained model, learned from large pretraining datasets. For a target application, some bases are advantageous while many others are not. To select the bases beneficial for the target application, we propose retraining the singular values (i.e., the weights of the bases) while keeping the bases fixed, using the training set of the target application. After retraining, we prune the bases associated with small singular values, as they are less important for the target application, and retain only those with large singular values, which are most critical for the target application. This approach allows us to eliminate the redundant parts of the original LLMs and retain only the components essential for the target application. To handle the data distribution differences between the pretraining dataset and the target application, we also augment the model with new bases learned from the training set of the target application during the pruning process. This enables us to learn the new bases necessary for the target application that are absent in the pretrained model.

We evaluate Basel on two models—Llama 2-7B and Llama 2-13B (Touvron et al., 2023)—and two tasks—mathematical reasoning and code generation. We evaluate the pass@1 accuracy of the compressed models on GSM8K (Cobbe et al., 2021) and MATH (Hendrycks et al., 2021) for the mathematical reasoning task and HumanEval (Chen et al., 2021a) and MBPP (Austin et al., 2021) for the code generation task. Compared to state-of-the-art baselines, our approach achieves substantially better performance, improving accuracy by up to 16% when the compression ratio [1] exceeds 6 for mathematical reasoning and 4 for code generation. This also indicates that, in cases of deep compression, our method reduces model size by up to 1.7 times while maintaining comparable accuracy to baseline methods.

This paper makes the following critical contributions:

- We analyze the relationship between pretrained models and target applications, highlighting the opportunity and underlying rationale for using low-rank decomposition to compress large language models while maintaining performance on target applications.

- We propose Basel, a low-rank decomposition approach to compress pretrained large language models for target applications. Basel identifies the beneficial and redundant components of large language models by relearning the importance (i.e., singular values) of bases using the training set of the target application, and then selects bases based on their importance.

- We evaluate Basel across multiple tasks and models, demonstrating its superior performance in deep compression.

---

[1]The compression ratio is defined as the ratio of the original model size to the compressed model size.

## 2 RELATED WORK

Singular Value Decomposition (SVD) (Golub & Van Loan, 1996) has been applied to reduce the size of machine learning models. Prior research (Xue et al., 2013; Jaderberg et al., 2014; Denton et al., 2014; Zhang et al., 2015; Povey et al., 2018; Chen et al., 2018; Acharya et al., 2019; Noach & Goldberg, 2020) has developed various SVD algorithms to compress different components of models, such as DNNs, CNNs, and embedding layers for a range of applications including natural language processing, speech, and vision. The primary distinction between our work and these prior studies is that they do not *relearn* the importance of bases using the training data of target applications. Instead, they typically prune bases according to the singular values from the original or finetuned models. FWSVD (Hsu et al., 2022) evaluates the importance of individual weights rather than bases during SVD. As highlighted in Sections 1 and 3, the bases hold significant physical meanings. Considering importance at this level of granularity results in improved performance. As shown in Section 4, our approach surpasses FWSVD in deep compression performance.

A recent study (Sharma et al., 2024) applied SVD to large language models. Its focus is on determining the optimal rank for each layer, while our emphasis is on basis selection. The two methods are orthogonal but complementary and can be combined. (Chen et al., 2021b) and (Yu & Wu, 2023) suggest reconstructing bases based on feature mimicking. These approaches are orthogonal to ours—they concentrate on *basis reconstruction*, whereas we focus on *basis selection*. Their methods complement ours and can be integrated together to achieve enhanced compression results.

## 3 BASEL

In this section, we describe our proposed compression method, Basel.

For a linear layer $\mathbf{y} = \mathbf{W}\mathbf{x} + \mathbf{b}$, Singular Value Decomposition (SVD) factorizes its weight matrix $\mathbf{W} \in \mathbb{R}^{n \times m}$ as the product of three matrices $\mathbf{U}$, $\mathbf{S}$, and $\mathbf{V}$:

$$\mathbf{W} = \mathbf{U}\mathbf{S}\mathbf{V}^{\mathrm{T}} \tag{1}$$

where $\mathbf{U} = [\mathbf{u}_1, \cdots, \mathbf{u}_r]$, $\mathbf{S} = \mathrm{diag}\,(s_1, \cdots, s_r)$, and $\mathbf{V} = [\mathbf{v}_1, \cdots, \mathbf{v}_r]$.

The values $\{s_i \in \mathbb{R}, i = 1, \cdots, r\}$ are positive singular values.[2] The vectors $\{\mathbf{u_i} \in \mathbb{R}^n, i = 1, \cdots, r\}$ and $\{\mathbf{v_i} \in \mathbb{R}^m, i = 1, \cdots, r\}$ are orthonormal, i.e., $\|\mathbf{u_i}\| = 1$, $\|\mathbf{v_i}\| = 1$, $\mathbf{u_i} \perp \mathbf{u_j}$, and $\mathbf{v_i} \perp \mathbf{v_j}$ if $i \neq j$.

Therefore, we can factorize matrix $W$ as the following series:

$$\mathbf{W} = \sum_{i=1}^{r} s_i \mathbf{u}_i \mathbf{v}_i^{\mathrm{T}} \tag{2}$$

Let matrix $\mathbf{W_i} = \mathbf{u}_i \mathbf{v}_i^{\mathrm{T}}$, then

$$\|\mathbf{W_i}\| = \sqrt{\mathrm{tr}\left(\mathbf{W_i}^{\mathrm{T}}\mathbf{W_i}\right)} = \sqrt{\mathrm{tr}\left(\mathbf{v_i}\mathbf{u_i}^{\mathrm{T}}\mathbf{u_i}\mathbf{v_i}^{\mathrm{T}}\right)} = \sqrt{\mathrm{tr}(\mathbf{v_i}\mathbf{v_i}^{\mathrm{T}})} = \sqrt{\mathrm{tr}(\mathbf{v_i}^{\mathrm{T}}\mathbf{v_i})} = 1 \tag{3}$$

$$\langle \mathbf{W_i}, \mathbf{W_j} \rangle = \mathrm{tr}\left(\mathbf{W_i}^{\mathrm{T}}\mathbf{W_j}\right) = \mathrm{tr}\left(\mathbf{v_i}\mathbf{u_i}^{\mathrm{T}}\mathbf{u_j}\mathbf{v_j}^{\mathrm{T}}\right) = 0, \quad \text{if } i \neq j \tag{4}$$

Therefore, $\left\{\mathbf{u}_i \mathbf{v}_i^{\mathrm{T}}, i = 1, \cdots, r\right\}$ can be seen as a group of orthonormal bases in a subspace of $\mathbb{R}^{n \times m}$, and $\{s_i, i = 1, \cdots, r\}$ are their weights, making the weight matrix $\mathbf{W}$ a linear combination of these bases.

This group of bases can be viewed as a series of filters that manipulate the input signal $\mathbf{x}$ to produce the output signal $\mathbf{y}$:

$$\mathbf{y} = \mathbf{W}\mathbf{x} + \mathbf{b} = \sum_{i=1}^{r} s_i \mathbf{u_i}\mathbf{v_i}^{\mathrm{T}}\mathbf{x} + \mathbf{b} = \sum_{i=1}^{r} s_i \langle \mathbf{x}, \mathbf{v_i} \rangle \mathbf{u_i} + \mathbf{b} \tag{5}$$

---

[2]We drop zero singular values and the corresponding columns of matrices $U$ and $V$.

In other words, for each basis (i.e., filter) $\mathbf{u}_i\mathbf{v_i}^\mathrm{T}$, the similarity between the input signal $\mathbf{x}$ and the unit direction vector $\mathbf{v_i}$ is measured by their inner product. This inner product is then multiplied by the (positive) singular value $s_i$ to determine the weight for the unit direction vector $\mathbf{u_i}$. The output signal $\mathbf{y}$ is the weighted sum of $\mathbf{u_i}$. Figure 2 illustrates this interpretation of the role of bases from the perspective of signal processing.

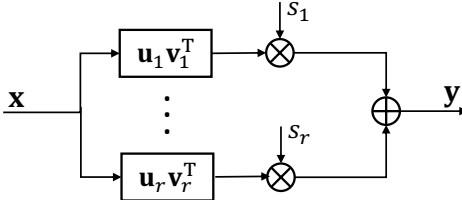

Figure 2: An interpretation of the role of bases from the perspective of signal processing.

Large language models pretrained on diverse datasets contain bases that capture a wide range of meanings. We factorize the $\mathbf{W_O^h}\mathbf{W_V^h}$ matrix from the attention layers in both the vanilla Llama 2-7B and the math-finetuned Llama 2-7B models, and decompose the $\mathbf{u}$ and $\mathbf{v}$ vectors associated with bases having large singular values. Table 1 presents our observations. First, some bases exhibit highly specific meanings, such as technology, programming, location, non-English characters, and math symbols. Second, within the same layer, different bases can encode vastly different meanings. For instance, in the math-finetuned model, layer 22, attention head 6, basis 1 corresponds to non-English characters, while basis 3 corresponds to math symbols. Third, fine-tuning does not automatically eliminate irrelevant bases. For example, the non-English character basis persists even after fine-tuning the model on math datasets.

These findings suggest that many bases in the model are useful for specific tasks, but may be irrelevant for others. When these irrelevant bases are used as filters in non-target applications, two scenarios can occur: the filter may not be activated (due to a small inner product $\langle \mathbf{x}, \mathbf{v}_i \rangle$), or worse, the filter is activated, introducing harmful information into the output and degrading performance. This indicates that pruning such bases could reduce model size with minimal performance loss, and in some cases, even enhance performance for the target application.

Table 1: The meaning of bases in Vanilla and math-finetuned Llama 2-7B

| Domain | Basis | Top ten most probable tokens corresponding to the basis |
|---|---|---|
| Technology | Vanilla model, Layer 16, Head 25, Basis 2 | _iOS, _Xcode, _ios, _Apple, _Mac, _iPhone, _app, _xcode, _NS, _App |
| Programming | Vanilla model, Layer 17, Head 6, Basis 1 | ., _in, <0x0A>, ..., _..., _and, L', _to, for, ! |
| Location | Vanilla model, Layer 17, Head 25, Basis 1 | _Massachusetts, _Illinois, _Chicago, _Boston, _Dan, _Harvard, _Connecticut, _IL, _Bulg, _Bulgar |
| Non-English | Math finetuning model, Layer 22, Head 6, Basis 1 | 學, 會, 區, 國, 經, 進, :, unk char, 無, 設 |
| Math | Math finetuning model, Layer 22, Head 6, Basis 3 | _{, {, }{, _{r, ={, _{`, _{", ]{, _`{, _{}; |

In our approach, Basel, we determine the importance of the bases from the pretrained model by retraining their singular values on the training set of the target application. The weight matrix $\widetilde{W}$ in Basel is represented as:

$$\widetilde{\mathbf{W}} = \sum_{i=1}^{r} \tilde{s}_i \mathbf{u_i}\mathbf{v_i}^\mathrm{T} + \sum_{j=1}^{\tilde{r}} \tilde{\mathbf{u}}_\mathbf{j}\tilde{\mathbf{v}}_\mathbf{j}^\mathrm{T} \tag{6}$$

In the first term, $\mathbf{u_i}\mathbf{v_i}^\mathrm{T}$ represents the original bases in the pretrained model. To assess their importance, we initialize their weights $\tilde{s}_i$ with their original singular values and then retrain these

weights (while keeping the bases fixed) on the training set of the target application. The aim is that, after retraining, the bases important for the target application will have larger singular values, whereas those that are useless or detrimental will have zero or very small singular values. This allows us to identify and prune the less useful bases. Relearning the importance of bases for the target application distinguishes our approach from previous methods. Prior approaches either use the singular values in the original model (Xue et al., 2013; Jaderberg et al., 2014; Denton et al., 2014; Zhang et al., 2015; Povey et al., 2018; Chen et al., 2018; Acharya et al., 2019; Noach & Goldberg, 2020; Sharma et al., 2024) or assess the importance of weight parameters, other than the importance of the bases, to prune them (Hsu et al., 2022). They do not relearn the importance of the bases specifically for the target application. From a signal processing perspective, this first term allows us to adjust the weight for each filter, catering to the needs of the target application.

In the second term, $\tilde{\mathbf{u}}_{\mathbf{j}}$ and $\tilde{\mathbf{v}}_{\mathbf{j}}$ are learnable vectors included for two primary purposes. First, due to differences in data distribution between the pretrained dataset and the target application, some bases necessary for the target application might be absent in the pretrained dataset. We use these vectors to learn such bases. Second, during pruning, although each pruned basis may individually have minimal impact on the target application, their cumulative performance loss can be significant. These additional vectors help compensate for the performance loss caused by the pruned bases. The number of learnable vectors $\tilde{r}$ is referred to as the *additional dimension*. From a signal processing perspective, this second term allows us to include additional, new filters to enhance the model performance on the target application.

---

**Algorithm 1:** Basel Algorithm

---

**Input:** Pretrained or Finetuning Model $M$
**Output:** Compressed Model $M'$
**Data:** KeepRatio, PruningTimes, KeepingEpoch, PruningEpoch, PostFineTuningEpoch

1. IterationsPerPruning = round(NumIterationsPerEpoch * PruningEpoch / PruningTimes);
2. KeepRatioPerPruning = KeepRatio$^{(1/\text{PruningTimes})}$;
3. Convert the weight matrix of each layer in $M$ into the form of equation equation 6;
4. **for** $i = 1$ **to** *KeepingEpoch* **do**
5.     Tune the learnable parameters in equation equation 6;
6. **end**
7. **for** $i = 1$ **to** *PruningEpoch* **do**
8.     Tune the learnable parameters;
9.     **if** *IterationID is a multiple of IterationsPerPruning* **then**
10.        **for** *each linear layer* **do**
11.           Prune bases with smaller singular values $\tilde{s}_i$ such that after pruning, the sum of the singular values of the remaining bases is KeepRatioPerPruning of the sum before pruning;
12.        **end**
13.     **end**
14. **end**
15. **for** *each layer* **do**
16.     Compute the low rank matrix $\widetilde{W}$ based on equation equation 6;
17.     $[U', S', V'] = \text{SVD}(\widetilde{W})$;
18.     Use two linear layers to substitute for the original layer;
19.     The first layer's weight matrix is $S'V'^{\text{T}}$;
20.     The second layer's weight matrix is $U'$;
21. **end**
22. **for** $i = 1$ **to** *PostFineTuningEpoch* **do**
23.     FineTune the new model $M'$;
24. **end**

---

Algorithm 1 outlines our approach. Basel takes a pretrained or fine-tuned model as input and gradually prunes bases in the original model. After each pruning step, it finetunes the learnable parameters $\tilde{s}_i$, $\tilde{\mathbf{u}}_j$, and $\tilde{\mathbf{v}}_j$ to offset the performance loss. Ultimately, a new weight matrix $\widetilde{\mathbf{W}}$ with a smaller rank $r'$ is learned. We perform a standard SVD on it, representing it as the product of matrices $\mathbf{U}'$, $\mathbf{S}'$,

and $\mathbf{V}'^{\mathrm{T}}$. We then replace the original layer with two new layers: $\mathbf{S}'\mathbf{V}'^{\mathrm{T}}$ becomes the weight matrix of the first new layer, and $\mathbf{U}'$ becomes the weight matrix of the second new layer. This reduces the number of parameters from $nm$ to $(n + m)r'$. The new model is subsequently further finetuned to enhance its performance on the target application.

## 4 EXPERIMENTS

### 4.1 EVALUATION METHODOLOGY

The performance of low-rank compression algorithms is evaluated on two tasks: mathematical reasoning and code generation. For each task, Llama 2-7B and Llama 2-13B models (Touvron et al., 2023) are first finetuned on a training dataset and then compressed using a compression algorithm. The compressed models are further finetuned before evaluation.

For the mathematical reasoning task, we utilize two evaluation datasets: GSM8K (Cobbe et al., 2021) and Hendrycks' MATH (Hendrycks et al., 2021). The GSM8K dataset comprises verbally described mathematical questions, containing 1,319 samples used for evaluation. The Hendrycks' MATH dataset covers more complex topics such as linear algebra and geometry, consisting of 5,000 question-answer pairs used for evaluation. Due to its complexity, the Hendrycks' MATH dataset necessitates more sophisticated computations and reasoning, resulting in lower accuracy compared to GSM8K.

For the code generation task, we use two evaluation datasets: MBPP (Austin et al., 2021) and HumanEval (Chen et al., 2021a). Both datasets evaluate the models' ability to generate Python code. MBPP comprises 500 code generation questions, while HumanEval includes 164 code generation questions.

We compare our proposed Basel with state-of-the-art low-rank compression algorithms, specifically SVD and FWSVD. SVD is widely used in previous model compression studies (Sharma et al., 2024; Acharya et al., 2019; Noach & Goldberg, 2020; Xue et al., 2013; Jaderberg et al., 2014; Denton et al., 2014; Zhang et al., 2015; Povey et al., 2018). FWSVD (Hsu et al., 2022) enhances SVD by evaluating the importance of model weight parameters.

### 4.2 RESULTS ON MATHEMATICAL REASONING

Figures 3(a) and (b) depict the performance of various compression algorithms on the Llama 2-7B model for the mathematical reasoning task. We evaluate the models' accuracy (Pass@1) at different compression ratios (original model size vs. compressed model size). For low compression ratios (below 6), all methods achieve similar accuracy. However, our Basel method significantly outperforms SVD and FWSVD at higher compression ratios. For instance, at a 7x compression ratio, Basel achieves around 43% and 10% accuracy on GSM8K and MATH datasets, respectively, while

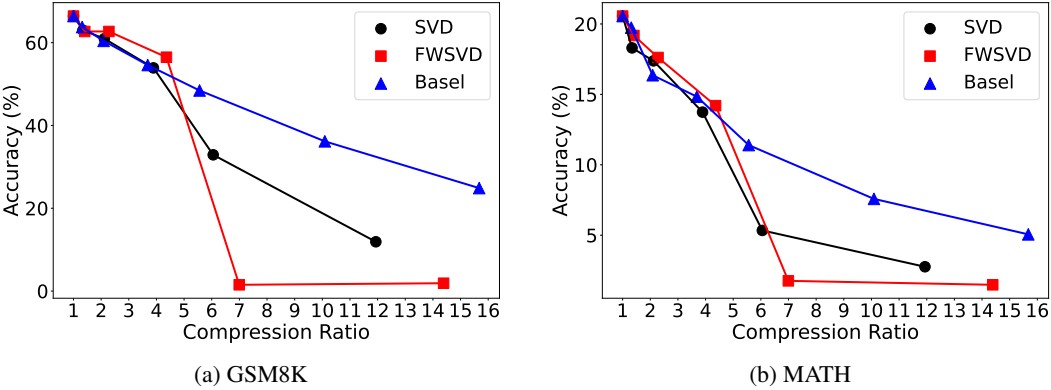

(a) GSM8K                    (b) MATH

Figure 3: Pass@1 accuracy and model size of Llama 2-7B compressed by various algorithms for the mathematical reasoning task (the datapoint values are provided in Table 2 of the appendix).

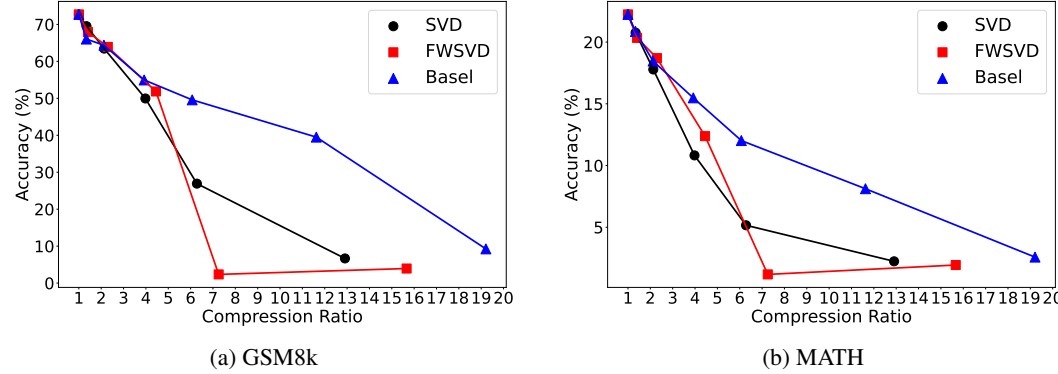

(a) GSM8k  (b) MATH

Figure 4: Pass@1 accuracy and model size of Llama 2-13B compressed by various algorithms for the mathematical reasoning task (the datapoint values are provided in Table 3 of the appendix).

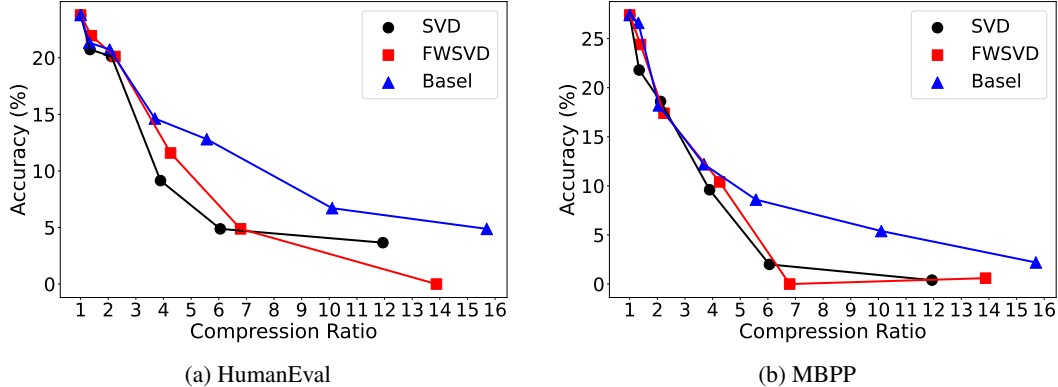

(a) HumanEval  (b) MBPP

Figure 5: Pass@1 accuracy and model size of Llama 2-7B compressed by various algorithms for the code generation task (the datapoint values are provided in Table 4 of the appendix).

FWSVD drops below 2% accuracy on both datasets and SVD reaches only 27% and 5% accuracy on GSM8K and MATH, respectively. We also find that Basel, at a compression ratio of 10, achieves better accuracy on both GSM8K and MATH compared to FWSVD and SVD at a compression ratio of 6. This suggests that Basel reduces the model size by up to 1.7 times more than the baseline methods while maintaining similar accuracy. This highlights the effectiveness of Basel for deep compression, especially when aiming for aggressive model size reduction.

Similar trends emerge for the larger Llama 2-13B model in Figures 4(a) and (b). Once again, Basel significantly outperforms SVD and FWSVD at compression ratios exceeding 6. At a 7x compression ratio, Basel achieves 47% and 11% accuracy on GSM8K and MATH datasets, respectively, demonstrating its advantage. This is in stark contrast to SVD's performance (23% and 5% accuracy on GSM8K and MATH) and FWSVD's near-complete accuracy drop (around 2% on both datasets). These results solidify Basel's effectiveness for deep compression across different model sizes.

### 4.3 RESULTS ON CODE GENERATION

Similar results extend to code generation tasks (Figures 5 and 6). For both Llama 2-7B and Llama 2-13B models, all methods perform comparably at lower compression ratios (below 4). However, Basel exhibits clear superiority at higher compression ratios. On Llama 2-7B at a 6x compression ratio, Basel achieves 12% and 8% accuracy on HumanEval and MBPP datasets, respectively, significantly outperforming SVD (5% and 2%) and FWSVD (6% and 2%). Similar trends hold for Llama 2-13B. These findings further solidify Basel's effectiveness for deep compression across diverse tasks and model sizes.

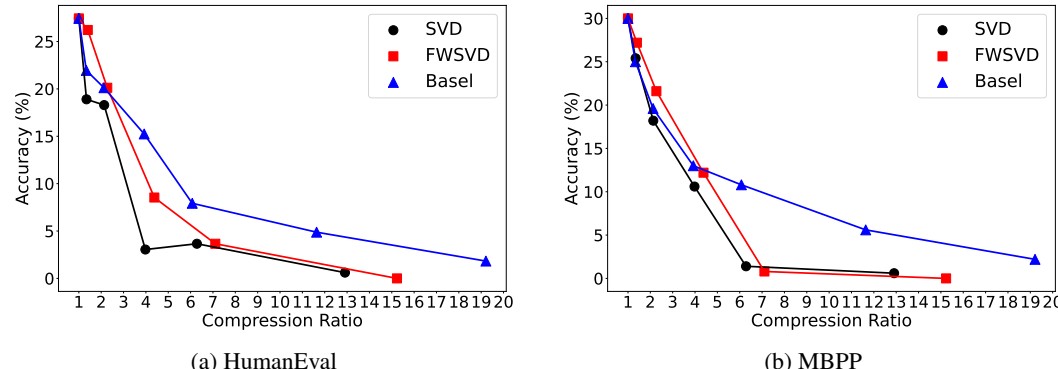

Figure 6: Pass@1 accuracy and model size of Llama 2-13B compressed by various algorithms for the code generation task (the datapoint values are provided in Table 5 of the appendix).

## 4.4 INFERENCE

Figure 7 presents the inference throughput and memory consumption of models compressed from Llama-7B on a single A100 GPU, using GSM8K as the evaluation set. The results show that low-rank compression methods, including SVD, FWSVD, and Basel, lead to reduced memory consumption and improved throughput as the model size decreases. Throughput and memory usage are primarily dependent on model size, with no significant differences between the methods at equivalent sizes. However, since our proposed Basel method achieves a greater reduction in model size while maintaining similar accuracy to SVD and FWSVD, it improves throughput by up to 16% and reduces memory consumption by up to 27%.

## 4.5 ABLATION STUDY

To analyze the impact of key parameters of Basel, we conducted ablation studies on the additional dimension (denoted by $\tilde{r}$ in equation equation 6) and pruning times. The additional dimension compensates for information loss during pruning, especially for deep compression. Figure 8 compares the performance of compressing Llama 2-7B for the mathematical reasoning task with and without an additional dimension of 32. As expected, incorporating an additional dimension improves accuracy at higher compression ratios (above 6). Similarly, pruning the model lightly but multiple times allows it to gradually adapt to the reduction in parameters after each pruning. This is particularly beneficial for achieving extreme compression ratios. Figure 9 demonstrates this effect by comparing the performance of pruning Llama 2-7B 100 times vs. 2 times on the same task. Here, pruning 100 times leads to better accuracy at compression ratios exceeding 10.

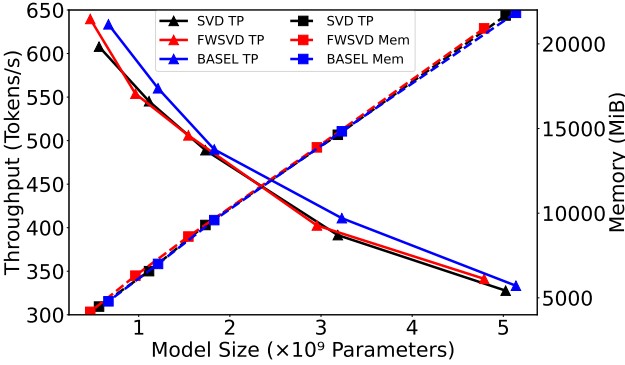

Figure 7: Throughput and memory consumption of compressed models.

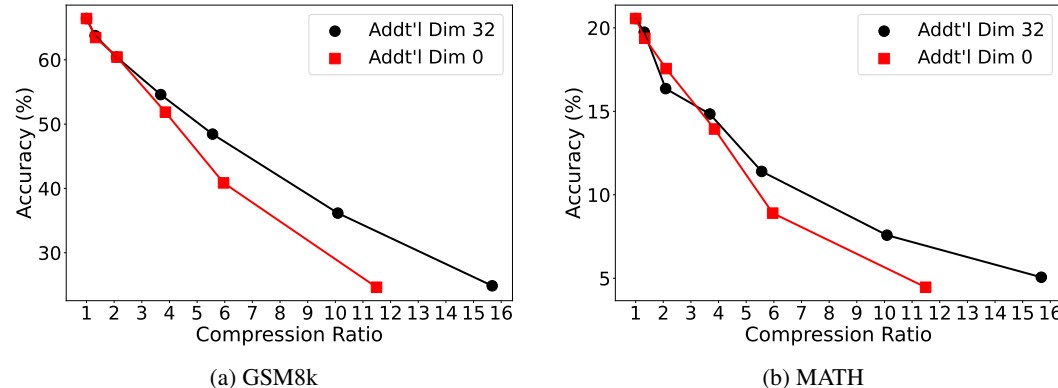

(a) GSM8k

(b) MATH

Figure 8: Ablation study: Effect of varying the additional dimension of Basel on compressing Llama 2-7B for the mathematical reasoning task (the datapoint values are provided in Table 6 of the appendix).

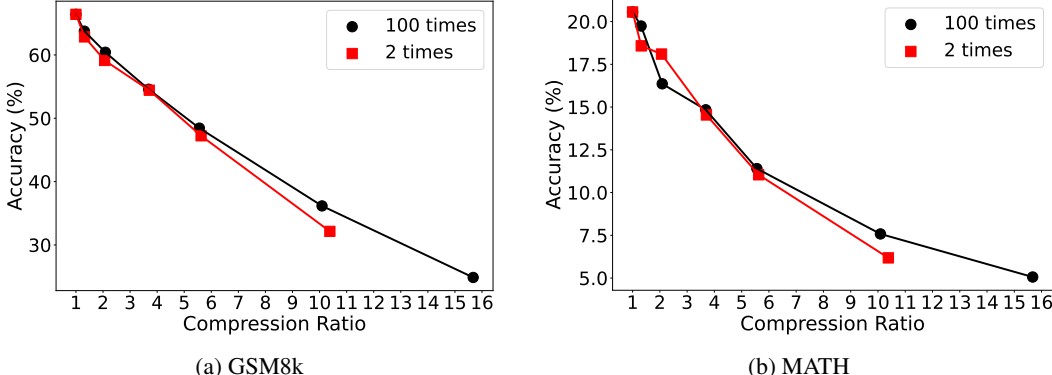

(a) GSM8k

(b) MATH

Figure 9: Ablation study: Effect of varying the pruning times of Basel on compressing Llama 2-7B for the mathematical reasoning task (the datapoint values are provided in Table 7 of the appendix).

## 5 CONCLUSION

The significant size of large language models leads to high inference costs and demands substantial computing resources. To mitigate these issues, we focus on compressing large language models to meet the specific requirements of target applications. Our approach involves examining these models through the lens of matrix factorization. By viewing the weight matrix of large language models as a linear combination of a group of bases, we have identified that pretrained models often contain many redundant bases that are less useful for target applications. To address this, we propose Basel, a compression algorithm that evaluates the importance of each base for target applications and prunes those that are less significant. Experimental results demonstrate that Basel significantly outperforms state-of-the-art low-rank compression algorithms in achieving deep compression. Basel greatly reduces the inference cost of large language models, making them more accessible and practical for a wider range of applications. This advancement has the potential to democratize the use of large language models, facilitating their adoption and integration across diverse fields and industries.

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

## A  APPENDIX

Table 2: Pass@1 accuracy and model size of Llama 2-7B compressed by various algorithms for the mathematical reasoning task.

| | | | | | | | |
|---|---|---|---|---|---|---|---|
| SVD | Model Size (B) | 6.74 | 5.02 | 3.18 | 1.73 | 1.11 | 0.56 |
| | GSM8K Acc (%) | 66.4 | 63.0 | 61.0 | 53.9 | 32.9 | 11.9 |
| | MATH Acc (%) | 20.6 | 18.3 | 17.4 | 13.7 | 5.3 | 2.8 |
| FWSVD | Model Size (B) | 6.74 | 4.79 | 2.95 | 1.54 | 0.96 | 0.47 |
| | GSM8K Acc (%) | 66.4 | 62.7 | 62.7 | 56.5 | 1.5 | 1.9 |
| | MATH Acc (%) | 20.6 | 19.2 | 17.6 | 14.2 | 1.8 | 1.5 |
| Basel | Model Size (B) | 6.74 | 5.14 | 3.23 | 1.83 | 1.21 | 0.67 | 0.43 |
| | GSM8K Acc (%) | 66.4 | 63.8 | 60.4 | 54.6 | 48.4 | 36.2 | 24.9 |
| | MATH Acc (%) | 20.6 | 19.7 | 16.4 | 14.8 | 11.4 | 7.6 | 5.1 |

Table 3: Pass@1 accuracy and model size of Llama 2-13B compressed by various algorithms for the mathematical reasoning task.

| | | | | | | | |
|---|---|---|---|---|---|---|---|
| SVD | Model Size (B) | 13.02 | 9.70 | 6.10 | 3.27 | 2.07 | 1.01 |
| | GSM8K Acc (%) | 72.7 | 69.5 | 63.5 | 50.0 | 26.9 | 6.7 |
| | MATH Acc (%) | 22.2 | 20.8 | 17.8 | 10.8 | 5.2 | 2.2 |
| FWSVD | Model Size (B) | 13.02 | 9.24 | 5.67 | 2.93 | 1.79 | 0.83 |
| | GSM8K Acc (%) | 72.7 | 67.9 | 63.9 | 51.9 | 2.4 | 3.9 |
| | MATH Acc (%) | 22.2 | 20.3 | 18.7 | 12.4 | 1.2 | 1.9 |
| Basel | Model Size (B) | 13.02 | 9.75 | 6.13 | 3.32 | 2.14 | 1.12 | 0.68 |
| | GSM8K Acc (%) | 72.7 | 66.0 | 64.4 | 55.0 | 49.6 | 39.5 | 9.2 |
| | MATH Acc (%) | 22.2 | 20.9 | 18.5 | 15.5 | 12.0 | 8.1 | 2.6 |

Table 4: Pass@1 accuracy and model size of Llama 2-7B compressed by various algorithms for the code generation task.

| | | | | | | | |
|---|---|---|---|---|---|---|---|
| SVD | Model Size (B) | 6.74 | 5.02 | 3.18 | 1.73 | 1.11 | 0.56 |
| | HumanEval Acc (%) | 23.8 | 20.7 | 20.1 | 9.1 | 4.9 | 3.7 |
| | MBPP Acc (%) | 27.4 | 21.8 | 18.6 | 9.6 | 2.0 | 0.4 |
| FWSVD | Model Size (B) | 6.74 | 4.84 | 3.01 | 1.58 | 0.99 | 0.49 |
| | HumanEval Acc (%) | 23.8 | 22.0 | 20.1 | 11.6 | 4.9 | 0 |
| | MBPP Acc (%) | 27.4 | 24.4 | 17.4 | 10.4 | 0 | 0.6 |
| Basel | Model Size (B) | 6.74 | 5.14 | 3.28 | 1.83 | 1.21 | 0.67 | 0.43 |
| | HumanEval Acc (%) | 23.8 | 21.3 | 20.7 | 14.6 | 12.8 | 6.7 | 4.9 |
| | MBPP Acc (%) | 27.4 | 26.6 | 18.2 | 12.2 | 8.6 | 5.4 | 2.2 |

Table 5: Pass@1 accuracy and model size of Llama 2-13B compressed by various algorithms for the code generation task.

| | Model Size (B) | 13.02 | 9.70 | 6.10 | 3.27 | 2.07 | 1.01 | |
|---|---|---|---|---|---|---|---|---|
| SVD | | | | | | | | |
| | HumanEval Acc (%) | 27.4 | 18.9 | 18.3 | 3.0 | 3.7 | 0.6 | |
| | MBPP Acc (%) | 30.0 | 25.4 | 18.2 | 10.6 | 1.4 | 0.6 | |
| | Model Size (B) | 13.02 | 9.31 | 5.73 | 2.97 | 1.83 | 0.85 | |
| FWSVD | | | | | | | | |
| | HumanEval Acc (%) | 27.4 | 26.2 | 20.1 | 8.5 | 3.7 | 0 | |
| | MBPP Acc (%) | 30.0 | 27.2 | 21.6 | 12.2 | 0.8 | 0 | |
| | Model Size (B) | 13.02 | 9.75 | 6.13 | 3.32 | 2.14 | 1.12 | 0.68 |
| Basel | | | | | | | | |
| | HumanEval Acc (%) | 27.4 | 22.0 | 20.1 | 15.2 | 7.9 | 4.9 | 1.8 |
| | MBPP Acc (%) | 30.0 | 25.0 | 19.6 | 13.0 | 10.8 | 5.6 | 2.2 |

Table 6: Ablation study: Effect of varying the additional dimension of Basel on compressing Llama 2-7B for the mathematical reasoning task.

| | Model Size (B) | 6.74 | 5.14 | 3.23 | 1.83 | 1.21 | 0.67 | 0.43 |
|---|---|---|---|---|---|---|---|---|
| Addt'l Dim 32 | | | | | | | | |
| | GSM8K Acc (%) | 66.4 | 63.8 | 60.4 | 54.6 | 48.4 | 36.2 | 24.9 |
| | MATH Acc (%) | 20.6 | 19.7 | 16.4 | 14.8 | 11.4 | 7.6 | 5.1 |
| | Model Size (B) | 6.74 | 5.06 | 3.20 | 1.75 | 1.13 | 0.59 | |
| Addt'l Dim 0 | | | | | | | | |
| | GSM8K Acc (%) | 66.4 | 63.5 | 60.4 | 51.9 | 40.9 | 24.6 | |
| | MATH Acc (%) | 20.6 | 19.4 | 17.6 | 13.9 | 8.9 | 4.5 | |

Table 7: Ablation study: Effect of varying the pruning times of Basel on compressing Llama 2-7B for the mathematical reasoning task.

| | Model Size (B) | 6.74 | 5.14 | 3.23 | 1.83 | 1.21 | 0.67 | 0.43 |
|---|---|---|---|---|---|---|---|---|
| 100 times | | | | | | | | |
| | GSM8K Acc (%) | 66.4 | 63.8 | 60.4 | 54.6 | 48.4 | 36.2 | 24.9 |
| | MATH Acc (%) | 20.6 | 19.7 | 16.4 | 14.8 | 11.4 | 7.6 | 5.1 |
| | Model Size (B) | 6.74 | 5.11 | 3.27 | 1.82 | 1.20 | 0.65 | |
| 2 times | | | | | | | | |
| | GSM8K Acc (%) | 66.4 | 62.9 | 59.1 | 54.4 | 47.2 | 32.1 | |
| | MATH Acc (%) | 20.6 | 18.6 | 18.1 | 14.5 | 11.0 | 6.2 | |

