# OpenReview forum: "Basel: Target-Aware Basis Selection for Language Models"
_ICLR.cc/2025/Conference — ICLR 2025 Conference Withdrawn Submission_

### Official Review · Reviewer_eEzr · 2024-10-20

**Soundness:** 3
**Presentation:** 3
**Contribution:** 2
**Rating:** 5
**Confidence:** 4

**Summary:**

The authors propose Basel, a target-aware low-rank model pruning and adaptive selection technique for large language models. Basel shows benefits over singular vector decomposition as it provides higher model performance at higher compression rates.

**Strengths:**

- Developing approaches for tailoring LLM applications to specific use cases is a highly relevant topic, as LLMs frequently offer much more capability than needed for downstream tasks. It also has a positive effect on the energy consumption and processing latency (i.e., end-user experience)
- The experimental design reflects a careful choice of models and datasets from different text domains.
- The manuscript is well-structured and easy to follow.

**Weaknesses:**

- Since the approach is iterative and gradually removes less relevant parameters, I would expect to see experimental results on how long it takes to adapt a model for the datasets. This could yield important insights on how expensive the Basel technique is.
- I would appreciate comparing performance and cost to well-established techniques such as knowledge distillation. I understand that, depending on the complexity of any given dataset, the cost may vary, especially with regard to retraining of the base (l. 209).
- The authors use reasoning benchmarks for their experiments only. What about language understanding (e.g., MMLU)? I would expect the language understanding capabilities of models to decrease, but at what point would this become noticeable?

**Questions:**

Please see the weaknesses.

---

> ### Author Response · Authors · 2024-12-02
>
> > Since the approach is iterative and gradually removes less relevant parameters, I would expect to see experimental results on how long it takes to adapt a model for the datasets. This could yield important insights on how expensive the Basel technique is.
> The computational overhead of Basel is comparable to baseline methods, such as SVD and FWSVD. To illustrate, consider compressing the 7B Llama model for a mathematical reasoning task:
>
> The computational overhead of Basel is comparable to baseline methods, such as SVD and FWSVD. To illustrate, consider compressing the 7B Llama model for a mathematical reasoning task:
>
> - **SVD** requires 3 epochs of fine-tuning, followed by compression, and then 3 epochs of post-compression fine-tuning, totaling 104-128 GPU hours.
> - **FWSVD** requires 3 epochs of fine-tuning, 1 epoch for profiling statistics (which is quite slow), compression, and then 3 epochs of post-compression fine-tuning, totaling 152-176 GPU hours.
> - Our method **Basel**, though initially stated to require a fine-tuned model as input, can effectively use a pre-trained model without performance loss. It involves 3 epochs for compression and 3 epochs for post-compression fine-tuning, totaling 108-132 GPU hours.
>
> Thus, the computational overhead of Basel is within the range of the baseline methods.
>
> &nbsp;
>
> > I would appreciate comparing performance and cost to well-established techniques such as knowledge distillation. I understand that, depending on the complexity of any given dataset, the cost may vary, especially with regard to retraining of the base (l. 209).
>
> While knowledge distillation and low-rank approximation both aim to improve model efficiency, they belong to distinct categories of model compression, each targeting different aspects of optimization. For example, knowledge distillation aligns activation distributions, while low-rank approximation focuses on compressing weight matrices into compact, low-rank forms.
>
> Because these methods address different challenges, direct comparisons may not provide a meaningful evaluation of their respective strengths. That said, techniques from these categories are often complementary and could potentially be combined to achieve even greater efficiency gains. For this paper, we chose to focus exclusively on low-rank approximation to ensure a rigorous and in-depth exploration of its potential.
>
> &nbsp;
>
> > The authors use reasoning benchmarks for their experiments only. What about language understanding (e.g., MMLU)? I would expect the language understanding capabilities of models to decrease, but at what point would this become noticeable?
>
> Our approach is not dependent on the specifics of reasoning or non-reasoning benchmarks, making it applicable to both types of tasks. Furthermore, the benchmarks we use—code generation and math question answering—also require a strong foundation in language understanding. Testing on these benchmarks effectively evaluates our approach's performance in language understanding, as providing correct answers inherently requires a solid grasp of the questions posed.

---

### Official Review · Reviewer_k4Yz · 2024-10-28

**Soundness:** 2
**Presentation:** 2
**Contribution:** 2
**Rating:** 3
**Confidence:** 4

**Summary:**

In this submission, the authors propose to compress LLMs for specific tasks by distinguishing between important and irrelevant parameters. The proposed method Basel, which is based on the main idea of SVD, aims to learn the scales of decomposed singular values and to learn additional bases from task-specific datasets. Experiments on math problem and code generation benchmarks demonstrate that Basel achieves better performance compared to SVD and FWSVD.

**Strengths:**

- The studied problem, i.e., reducing the computational costs of LLMs for specific tasks, is practical and important.
- The authors provide detailed descriptions of the proposed method, including background information, motivational insights, and algorithm process.

**Weaknesses:**

- The novelty of the proposed method is limited. As mentioned by the authors, there are existing works on utilizing singular values in LLMs and pruning based on importance. The technical contributions of this submission are not clear.
- The experiments are not convincing and lack some detail. (a) The training details of Basel are not provided. (b) Although the authors include lots of related works, they compare the proposed method against a very limited set of baselines. (c) Since Basel includes new trainable parameters (such as scales and additional bases) and is trained on task-specific datasets, it is not surprising that Basel can achieve better performance on these tasks. The conducted comparisons do not convincingly demonstrate the effectiveness of the proposed method. (d) The observations presented in Table 1 are interesting. The authors can consider providing similar observations for the LLMs trained using Basel to provide empirical evidence that it successfully identifies beneficial and redundant components.
- The writing of this submission should be further improved.

**Questions:**

Please refer to the Weaknesses above.

---

> ### Author Response · Authors · 2024-12-02
>
> > The novelty of the proposed method is limited. As mentioned by the authors, there are existing works on utilizing singular values in LLMs and pruning based on importance. The technical contributions of this submission are not clear.
>
> We respectfully disagree and highlight the following novel contributions of our work:
> - We identify the semantic meanings of bases in LLMs, providing new insights into their interpretability.
> - We propose a novel approach to relearn the importance of bases for target applications, rather than simply copying the importance from pretrained or fine-tuned models.
>
> As stated in the related works section (lines 115–117), prior studies prune bases based on singular values derived from pretrained or fine-tuned models without leveraging the training data of the target applications. In contrast, our method explicitly learns the importance of bases using task-specific data, enabling better adaptation to target tasks. This distinction represents a key technical innovation of our approach.
>
> &nbsp;
>
> >The experiments are not convincing and lack some detail. Although the authors include lots of related works, they compare the proposed method against a very limited set of baselines.
>
> We have compared our method against state-of-the-art low-rank decomposition techniques, which are the most relevant baselines for this work.
>
> &nbsp;
>
> > Since Basel includes new trainable parameters (such as scales and additional bases) and is trained on task-specific datasets, it is not surprising that Basel can achieve better performance on these tasks. The conducted comparisons do not convincingly demonstrate the effectiveness of the proposed method.
>
> We would like to clarify a few points:
> - The baseline approaches are also trained on task-specific datasets. The improved performance of Bazel is not merely a result of using task-specific training but rather stems from the effectiveness of our proposed method in optimizing base importance for target tasks.
> - When comparing our method with baseline approaches, we account for model size, including the new trainable parameters introduced by Bazel. Despite having a smaller overall model size, Bazel consistently achieves higher performance. This result underscores the efficiency of our method and is a notable achievement.
>
> We hope these clarifications address the concerns and highlight the effectiveness of our proposed approach.

---

### Official Review · Reviewer_Lbvs · 2024-10-30

**Soundness:** 2
**Presentation:** 3
**Contribution:** 2
**Rating:** 5
**Confidence:** 3

**Summary:**

The authors proposed a method called BASEL, which can streamline language models and reduce model size up to 1.7 times while maintaining similar accuracy.
The method is based on Singular Value Decomposition (SVD)  and can identify and remove unnecessary components from pre-trained models, keeping only the essential parts needed for specific tasks.

**Strengths:**

1. The paper is well written
2. The method can reduce the model size while keeping comparable accuracy on specific tasks.

**Weaknesses:**

## 1. Lack of comparison with other model compression methods
Currently, the most common compression method in practical is model quantization, but the experimental part of this paper does not compare with it. A set of comparative experiments with the state-of-art quantization method should be added.
On the other hand, model quantization is not task-specific, but Bazel can only be effective for specific tasks, so some additional discussion is needed to explain the necessity of this method.


## 2. Lack of experiments on the overhead of performing the proposed compression method

A set of experiments should be added to show the time and resource cost of performing Basel compression

## 3. Limited scope of application
One of the main advantages of LLM over previous models is its versatility. However, the method proposed in this paper reduces the size of the model by sacrificing the versatility of the model and only focuses on specific tasks. And the pruning of the model also requires
The author needs to give a reason for doing so.

**Questions:**

As described  in Weaknesses.

---

> ### Author Response · Authors · 2024-12-02
>
> > Lack of comparison with other model compression methods. Currently, the most common compression method in practical is model quantization, but the experimental part of this paper does not compare with it. A set of comparative experiments with the state-of-art quantization method should be added. On the other hand, model quantization is not task-specific, but Bazel can only be effective for specific tasks, so some additional discussion is needed to explain the necessity of this method.
>
> Low-rank approximation is a distinct category of model compression techniques, fundamentally different from approaches like quantization, weight pruning, and knowledge distillation. Each category leverages unique properties to improve model efficiency: quantization reduces weight precision, pruning exploits weight sparsity, knowledge distillation aligns activation distributions, and low-rank approximation compresses weight matrices into compact, low-rank forms. Since these methods target different aspects of model optimization, direct comparisons across categories are neither necessary nor meaningful. Moreover, techniques from different categories are often complementary and can be combined to achieve further efficiency gains. Therefore, this work focuses exclusively on low-rank approximation to ensure a focused and rigorous evaluation within this category.
> Regarding task specificity, model quantization can indeed be either task-specific or task-agnostic, depending on the application scenario. Bazel is task-specific, designed for situations where a pretrained model is fine-tuned and compressed for specific application domains. Such scenarios are common in practice; for instance, AR/VR companies often adapt large pretrained models for AR/VR-specific tasks.
>
> &nbsp;
>
> > Lack of experiments on the overhead of performing the proposed compression methodA set of experiments should be added to show the time and resource cost of performing Basel compression
>
> The computational overhead of Basel is comparable to baseline methods, such as SVD and FWSVD. To illustrate, consider compressing the 7B Llama model for a mathematical reasoning task:
>
> - **SVD** requires 3 epochs of fine-tuning, followed by compression, and then 3 epochs of post-compression fine-tuning, totaling 104-128 GPU hours.
> - **FWSVD** requires 3 epochs of fine-tuning, 1 epoch for profiling statistics (which is quite slow), compression, and then 3 epochs of post-compression fine-tuning, totaling 152-176 GPU hours.
> - Our method **Basel**, though initially stated to require a fine-tuned model as input, can effectively use a pre-trained model without performance loss. It involves 3 epochs for compression and 3 epochs for post-compression fine-tuning, totaling 108-132 GPU hours.
>
> Thus, the computational overhead of Basel is within the range of the baseline methods.
>
> &nbsp;
>
> > Limited scope of application. One of the main advantages of LLM over previous models is its versatility. However, the method proposed in this paper reduces the size of the model by sacrificing the versatility of the model and only focuses on specific tasks. And the pruning of the model also requires The author needs to give a reason for doing so.
>
> Versatility is valuable, but unnecessary versatility can be detrimental, as it introduces significant inference overhead. Bazel addresses this challenge by enabling the adaptation of a pretrained model for one or more specific application domains through compression. This approach is well-suited to practical scenarios, such as AR/VR companies that adapt large pretrained models to develop efficient, domain-specific solutions for their applications. By focusing on these targeted use cases, Bazel offers an effective method for optimizing LLMs for task-specific contexts without attempting to replicate general-purpose capabilities.

---

### Official Review · Reviewer_sUYG · 2024-11-03

**Soundness:** 3
**Presentation:** 2
**Contribution:** 2
**Rating:** 5
**Confidence:** 3

**Summary:**

This paper presents Basel, a new approach to compressing LLMs by identifying the importance of bases for a target application, pruning the others, and then finishing with a finetuning step. The approach is evaluated on two tasks, mathematical reasoning and code generation, and is compared to other SVD-based approaches.  The results show that for the llama2  model, Basel can achieve better performance than the other two approaches when the compression ratio exceeds 5.

**Strengths:**

Here are some of the paper's strengths:
1. Solid Motivation: Deploying LLMs on resource-constrained devices is a relevant problem, and attempting to address it through compression seems reasonable.
2. The authors outlined their approach very clearly.

**Weaknesses:**

The paper has a couple of weaknesses listed below:
1. Focusing on low-rank compression approaches for comparison: The paper does not compare other compression techniques widely used to compress LLMs, such as quantization and knowledge distillation. Authors could potentially mention them in related works and why they think they are not comparable.
2. Limited number of tasks and models: This paper could benefit from running more evaluations on different tasks other than mathematical reasoning and code generation. Additionally, it might be good to show that the approach is generalizable to other models of various sizes.
3. Some terms in the equations are not very well defined. For example, tr() in equation(3) is not defined.

**Questions:**

No questions for the authors.

---

> ### Author Response · Authors · 2024-12-02
>
> > Focusing on low-rank compression approaches for comparison: The paper does not compare other compression techniques widely used to compress LLMs, such as quantization and knowledge distillation. Authors could potentially mention them in related works and why they think they are not comparable.
>
> Low-rank approximation is a distinct category of model compression techniques, fundamentally different from approaches like quantization, weight pruning, and knowledge distillation. Each category leverages unique properties to improve model efficiency: quantization reduces weight precision, pruning exploits weight sparsity, knowledge distillation aligns activation distributions, and low-rank approximation compresses weight matrices into compact, low-rank forms. Since these methods target different aspects of model optimization, direct comparisons across categories are neither necessary nor meaningful. Moreover, techniques from different categories are often complementary and can be combined to achieve further efficiency gains. Therefore, this work focuses exclusively on low-rank approximation to ensure a focused and rigorous evaluation within this category.
>
> &nbsp;
>
> > Limited number of tasks and models: This paper could benefit from running more evaluations on different tasks other than mathematical reasoning and code generation. Additionally, it might be good to show that the approach is generalizable to other models of various sizes.
>
> Our paper includes evaluations on two distinct tasks—mathematical reasoning and code generation—spanning four datasets (GSM8K, MATH, HumanEval, and MBPP). Additionally, we test our approach on two models with different sizes, LLaMA-2 7B and LLaMA-2 13B, to assess its generalizability.
>
> While we understand the value of even broader evaluations, expanding experiments indefinitely is infeasible, as one could always request additional tasks or models. We hope the reviewer recognizes the diversity of our current evaluations and agrees that they sufficiently demonstrate the generalizability and effectiveness of our method.
>
> &nbsp;
>
> > Some terms in the equations are not very well defined. For example, tr() in equation(3) is not defined.
>
> The term tr(), used in Equation (3), denotes the trace of a matrix, which is the sum of its diagonal elements. This is a fundamental concept in linear algebra.

---

### Note · Authors · 2024-12-02

I have read and agree with the venue's withdrawal policy on behalf of myself and my co-authors.